# Genus-Level Analysis of Gut Microbiota in Children with Autism Spectrum Disorder: A Mini Review

**DOI:** 10.3390/children10071103

**Published:** 2023-06-23

**Authors:** Mariya Levkova, Trifon Chervenkov, Rouzha Pancheva

**Affiliations:** 1Department of Medical Genetics, Medical University Varna, Marin Drinov Str 55, 9000 Varna, Bulgaria; 2Laboratory of Medical Genetics, St. Marina Hospital, Hristo Smirnenski Blv 1, 9000 Varna, Bulgaria; 3Laboratory of Clinical Immunology, St. Marina Hospital, Hristo Smirnenski Blv 1, 9000 Varna, Bulgaria; 4Department of Hygiene and Epidemiology, Medical University Varna, Marin Drinov Str 55, 9000 Varna, Bulgaria

**Keywords:** gut microbiome, autism, autism spectrum disorder, gut-brain axis

## Abstract

Autism is a global health problem, probably due to a combination of genetic and environmental factors. There is emerging data that the gut microbiome of autistic children differs from the one of typically developing children and it is important to know which bacterial genera may be related to autism. We searched different databases using specific keywords and inclusion criteria and identified the top ten bacterial genera from the selected articles that were significantly different between the studied patients and control subjects studied. A total of 34 studies that met the inclusion criteria were identified. The genera *Bacteroides*, *Bifidobacterium*, *Clostridium*, *Coprococcus*, *Faecalibacterium*, *Lachnospira*, *Prevotella*, *Ruminococcus*, *Streptococcus*, and *Blautia* exhibited the most substantial data indicating that their fluctuations in the gastrointestinal tract could be linked to the etiology of autism. It is probable that autism symptoms are influenced by both increased levels of harmful bacteria and decreased levels of beneficial bacteria. Interestingly, these genera demonstrated varying patterns of increased or decreased levels across different articles. To validate and eliminate the sources of this fluctuation, further research is needed. Consequently, future investigations on the causes of autism should prioritize the examination of the bacterial genera discussed in this publication.

## 1. Introduction

Autism spectrum disorder (ASD) is defined as a neurological developmental disorder which is characterized by a broad range of symptoms, including impaired social interactions, repetitive behavior, and restricted interests and activities [1].

In many cases, the etiology of ASD is still unknown despite extensive research in recent years [2], although there have been studies on the possible roles of genetic variants, epigenetic modifications, imbalances of different neurotransmitters, oxidative stress, etc. However, only approximately 40% of the causes of ASD have been identified as symptoms of single gene and chromosomal disorders [2]. Moreover, no particular gene has been identified as the cause of ASD, and there are around 800 genes that have been associated with this disorder [2]. ASD can also be described as a multifactorial disorder due to genetic and environmental factors [2]. Recently, there has been growing interest in the relationship between ASD and the gut microbiome and its potential significance as an etiological component of ASD.

The so-called gut-brain axis is a bidirectional system that serves as a communication network between the central nervous system (CNS) and the gut microbiome (Figure 1). Different microbial products, metabolites, and debris may enter the bloodstream, reach the CNS, and influence it. In addition, there may be increased levels of cytokines in the plasma as a result of dysbiosis and an immune reaction [3,4]. Moreover, some types of bacteria are believed to be able to stimulate the vagus nerve by producing neuroactive molecules [5].

There are data that indicate the gut microbiome of autistic children is different from that of typically developing children [6]. It has been reported that patients with ASD had an abnormal composition of the gut microbiome, i.e., decreased levels of Bacteroides, Bifidobacterium, and Enterococcus, while Lactobacillus, Ruminococcus, and Escherichia-Shigella were increased [2]. The majority of research conducted so far has been observational; thus, it has been unable to conclusively show that the gut microbiome plays a part in the etiology of ASD. However, it may serve as a potential noninvasive biomarker for screening of children or could be the target of new therapies, such as fecal transplantation [7]. Given the significance of the gut microbiome, it is worth investigating which bacterial populations are noticeably different in autistic children and may be the focus of future experimental research. The purpose of this mini review is to list the bacterial genera for which there is the greatest evidence of a substantial difference between patients with ASD and typically developing children.

## 2. Material and Methods

Following the PRISMA guideline, a search was conducted, during February 2023, of several databases, i.e., PubMed, Web of Science, and Scopus [8]. We used key words that included gut, microbiota, and microbiome in combination with autism or autistic disorder. We did not apply any restrictions regarding the publication date. We used the following inclusion criteria: (1) Include case-control studies on the gut microbiome in autistic and normally developing children. (2) Exclude studies that focus only on animal models. (3) Included participants must be under 18 years old. (4) Control patients must not be related to index patients. (5) 16sRNA sequencing or whole metagenome sequencing must be used to identify the microbiome of the participants. (6) Full details about the significantly different genera must be available. (7) Only studies published in English and accessible in their entirety are considered for inclusion. Studies that did not meet the inclusion criteria were excluded from further analysis.

In order to identify the top ten genera that were reported most frequently in the articles as significantly different, either decreased levels or increased levels in the gut, the information that was available about the reported genera in the included studies was summarized in a table and used to perform a crosstab analysis (SPSS, version 26, IBM, Endicott, NY, USA).

## 3. Results

After completing the search, a total of 2688 studies were obtained from PubMed, 3606 studies from Web of Science, and 2158 studies from Scopus (Figure 2). Then, we detected duplicate studies and checked the studies for eligibility based on their titles and abstracts, excluding those that did not meet the inclusion criteria. In total, there were 34 studies that met the inclusion criteria. Information about the included studies is presented in Appendix A. The top ten genera, which were reported most times in the articles as significantly different, either decreased or increased levels in the gut, are listed below.

### 3.1. Bacteroides

Species from the genus *Bacteroides* belong to the group of Gram-negative bacteria and are among the first to colonize the gastrointestinal tract (GIT). They are widely found in GITs of all ages and are reported to be the main propionate producer, which is later used for gluconeogenesis in the liver [9]. The levels of propionic acid were reported to be higher in fecal samples from children with ASD [10]. This, taken together with the fact that the administration of propionic acid in animal models leads to similar ASD behavior [11], could explain why its production by *Bacteroides* could play a role in the etiology of ASD. Additionally, *Bacteroides* participate in the metabolism of tryptophan into indole in the GIT. Later, indole could be a substrate for the production of other metabolites in the liver, which have potential cytotoxic effects at high concentrations [12]. Moreover, tryptophan metabolites from the gut microbiome could activate the immune system, thus stimulating the gut-brain axis in a third way [12]. In addition, there are theories according to which if *Bacteroides* are reduced, this might lead to the so-called leaky gut, and microorganisms and their toxins might enter the blood stream [13].

*Bacteroides* were shown to be statistically different in 13 articles and were typically less common in people with ASD than in control participants, according to our findings [3,13,14,15,16,17,18,19,20,21,22,23,24]. However, there are conflicting results on the potential role of this genus. Two of the published meta-analyses stated that *Bacteroides* were more prevalent among children with ASD [25,26], while other studies claimed they were lower [27,28]. The contradictory findings could be explained by differences among the studied patients. Constipation or diarrhea have been shown to influence the amount of *Bacteroides* [28]. These conditions are common among ASD patients, and if there were not strict enough exclusion criteria, they could have interfered with the results of the gut composition. Therefore, additional case-control studies without patients with gastrointestinal dysfunction are required to determine the role of *Bacteroides*.

### 3.2. Bifidobacterium

Species from the genus *Bifidobacterium* are Gram-positive bacteria, and their presence is considered to be protective because of their production of indole-3-lactic acid. The latter suppresses the growth of Escherichia coli and other pathogens in the GIT and could prevent inflammation in the intestinal epithelium [29]. Furthermore, supplementation with probiotics rich in *Bifidobacterium* has been shown to improve the symptoms of ASD [27]. Due to this protective function and potential interaction with the host immune system and the gut-brain axis, this genus could be tested in children with ASD.

The results of our review show that *Bifidobacterium* was significantly different in a total of 13 studies [3,14,16,17,21,22,24,30,31,32,33,34,35], and decreased levels were reported in the majority of the tested subjects with ASD. Our results were in agreement with other publications, which also reported a reduction in *Bifidobacterium* [25,26,27]. However, there were also case-control studies that reported the opposite finding. This surprising result could be partially explained by different diets. For example, in a study from India, where *Bifidobacterium* levels increased, the ASD children were put on a gluten-free diet, which, according to the authors, had a positive effect on this genus [30]. In addition, there are data that vitamin supplementation has been shown to potentially reduce the abundance of *Bifidobacterium*, and there appears to be a negative correlation between the presence of this genus and age progression [21,31]. For example, one of the studies, which reported increased *Bifidobacterium* in ASD patients, had a very small sample size and a wide range of ages, which were both possible limitations of the study and could explain the surprising finding [35]. While most of the included studies adhered to strict inclusion criteria and excluded participants who had taken probiotics, there was no specific mention of vitamin supplementation in the reviewed literature. All of the above demonstrates that the gut microbiome is extremely sensitive to external influences and is rapidly changing.

### 3.3. Clostridium

*Clostridium* is another genus of Gram-positive bacteria that have been reported to be significantly increased in ASD patients [27]. Species belonging to the genus *Clostridium* produce different active molecules, i.e., alpha, beta, epsilon, iota toxins, and enterotoxins. The ability of this genus to synthesize neurotoxins and secrete them in the blood stream might contribute to ASD. Moreover, beta toxins have been reported to be significantly increased in fecal samples from autistic children [36]. In addition, toxin B, which is produced by *Clostridium difficile*, could lead to apoptosis of certain types of neurons and structural changes, such as an increased number of dendritic spines [37]. Thus, the increase in the percentage of *Clostridium* in children with ASD is an important question.

*Clostridium* was reported to be significantly different in eight studies, and it was more abundant in ASD patients than in the controls in six of the studies [3,6,13,14,18,31,38,39]. Our results were in agreement with previous meta-analyses, which also reported an increased percentage of this genus in children with ASD [25,27,37]. The geographical location of the studies could not explain the reported dissimilarities, since the studies from China reported both increased and decreased levels of *Clostridium*. However, a different diet could have a role in the presence of *Clostridium* because high intakes of carbohydrates and zinc seem to increase the abundance of *Clostridium* [40]. In addition, the diet of the mother could play a role in the microbiome composition of the child. For vaginally delivered children, increased intake of dairy products and fruits during pregnancy has been reported to be positively correlated with the presence of *Clostridium* in the infant’s intestine [41]. However, an increased percentage of *Clostridium* led to a higher concentration of propionic acid, which could play a role in the etiology of ASD [36]. In addition, there could be decreased concentrations due to the production of p-cresol by some species, part of the genus *Clostridium*. Glutathione is an antioxidant that has been linked to ASD [9]. This is why the abundant presence of *Clostridium* in the GIT of ASD patients remains an important research question regarding the etiology of this disorder. Furthermore, the production of several toxins by Clostridium raises the question of whether the toxins themselves, rather than the total number of this genus, have a role in ASD. Therefore, the amount of toxins produced by various bacterial genera should be measured as part of a functional testing of stool samples.

### 3.4. Coprococcus

*Coprococcus* is a genus of Gram-positive cocci which belong to the group of fermenting bacteria and produce butyrate [42]. This genus was reported in a total of eight of the studies as significantly different between ASD patients and neurotypical children, and its abundance was decreased in five of the included studies [19,20,24,35,42,43,44,45]. There is scarce literature on the role of *Coprococcus* in the etiology of ASD, as only one meta-analysis found a significantly lower percentage of this genus in ASD children [25]. Nevertheless, *Coprococcus* was reported to be correlated with stereotyped and anxiety-like behavior [46]. This could be explained by the butyrate production typical for the species in the genus *Coprococcus*. Butyrate acts as an anti-inflammatory agent by inhibiting the action of nuclear factor κB. It is also involved in reducing the production of reactive oxygen species, cell proliferation and differentiation, maintaining the protective barrier function of the intestinal mucosa, and intestinal motility [47]. Since a decreased percentage of *Coprococcus* would also lead to less butyrate synthesized in the GIT, one could hypothesize that this would interfere with the normal function of the immune system and might contribute to ASD by acting on the gut-brain axis. This is why *Coprococcus* might be considered to be a promising target for future research.

### 3.5. Faecalibacterium

*Faecalibacterium* is a genus of Gram-negative bacteria, and there is only one species, that belongs to this genus, i.e., *Faecalibacterium* prausnitzii. It was believed that *Faecalibacterium* has a commensal relationship with its host in the human GIT [48]. However, new data have shown that this genus was increased in patients with Crohn’s disease, and a higher abundance of *Faecalibacterium* could lead to inflammation in the GIT [48]. This could be due to the reported potential involvement of this genus in the control and regulation of genes responsible for the expression of interferon gamma [25]. Increased amounts of this cytokine could lead to an abnormal immune reaction in the host due to activation of the type I interferon signaling pathway [27,48].

According to the findings of our review, *Faecalibacterium* was significantly different in nine studies, and it was more frequently elevated in ASD patients than in control individuals [6,13,14,20,31,33,34,43,48]. This was in agreement with other studies, which also reported an increased abundance of this genus among autistic children [25,27]. This could be due to the altered expression of genes, which control the production of interferon gamma. The last one is hypothesized to be involved in neuroplasticity and the formation of synapses [25]. Moreover, it has been reported that the levels of interferon regulating factors 7 and 9 were closely associated with the amount of *Faecalibacterium* in the GIT. These two factors are also involved in the activation of the type I interferon signaling pathway [27]. Therefore, changes in the normal population of the gut microbiome might modulate the gut-brain axis by increasing concentrations of different cytokines, and therefore have an impact on autism spectrum disorders.

### 3.6. Lachnospira

*Lachnospira* is a genus of Gram-positive bacteria that produce the short-chain fatty acid (SCFA) butyrate, similar to the genus *Coprococcus* [26,42]. The role of the *Lachnospira* genus in the etiology of autism spectrum disorder (ASD) has yielded conflicting findings. On the one hand, lower levels of *Lachnospira* have been observed in individuals with generalized anxiety disorder [49]. Although anxiety is a distinct disorder, it is noteworthy that individuals with autism sometimes experience anxiety symptoms, suggesting potential interconnected etiological mechanisms. On the other hand, an increased abundance of *Lachnospira* could lead to elevated production of short-chain fatty acids (SCFAs) such as butyrate. Butyrate is known to impact cell signaling, neurotransmitters, free radicals, and immune system functions [11]. Thus, it is plausible that the gut-brain axis could be influenced by *Lachnospira*, whether its abundance is decreased or increased in the gastrointestinal tract (GIT).

The results of the present review were also uncertain. The genus was significantly different in eight studies, but the number of times its abundance was increased or decreased was the same [13,15,16,18,35,44,50,51]. Only one meta-analysis addressed the presence of *Lachnospira* in the GIT of autistic children, but the results were not significant [26]. However, an abnormal amount of *Lachnospira*, either increased or decreased, is supposed to have an effect on the concentration of butyrate. According to published data, butyrate can protect the intestinal epithelium, thus suppressing inflammation processes in the GIT. Moreover, butyrate is involved in the synthesis of dopamine by acting on the expression of the gene for tyrosine hydroxylase [44]. Therefore, it is a plausible explanation that a decrease in the amount of *Lachnospira* would also lead to a decrease in butyrate, which would have a negative impact on the gut-brain axis.

### 3.7. Prevotella

*Prevotella* is another example of Gram-negative bacteria that are normally found in the human GIT. The abundance of this genus depends on the type of diet—a diet rich in processed foods would lead to a decreased amount of *Prevotella* [52]. *Prevotella* is considered to be a beneficial bacterium because of its ability to degrade complex polysaccharides in the GIT. It also synthesizes propionate, similar to some of the genera described above [52]. However, there are also reports that *Prevotella* is associated with rheumatoid arthritis, particularly *Prevotella copri*. This species could stimulate the production of Prevotella copri-specific antibodies, which in turn could trigger an immune response from the host [52]. Therefore, at this time, the role of the *Prevotella* genus is not completely understood.

According to our results, *Prevotella* was significantly different in a total of eleven of the studies, and its abundance was decreased in the majority of them [17,18,20,32,34,35,39,42,43,44,53]. One meta-analysis investigated the role of *Prevotella* in ASD children, but it did not report significant results [26]. A systematic review from 2019 also reported lower levels of *Prevotella* among ASD children. However, in this case, the ASD patients also had gastrointestinal complaints such as constipation and irritable bowel disease [54]. Whether it is an increase or decrease in the genus *Prevotella* that is associated with ASD, remains unclear. If it is more abundant, it could lead to higher levels of propionic acid, which are linked to autism [11]. However, a species belonging to the genus *Prevotella* has been associated with an anti-inflammatory effect in GIT due to increased secretion of interleukin 10 [55]. Nevertheless, the decreased abundance of *Prevotella* could also be explained by the consumption of the so-called Western foods consumed by the included children with ASD [56]. Therefore, the role of the genus *Prevotella* remains to be an open question.

### 3.8. Ruminococcus

*Ruminococcus* is a genus of Gram-positive cocci that synthesize butyrate and are normally found in the GIT [27,51]. Some species belonging to this genus have been found to be more common in ASD children and have been associated with gastro-intestinal complaints in these patients [57]. Moreover, a member of this genus, *Ruminococcus gnavus*, has been reported to produce a polysaccharide called glucorhamnan, which could trigger an immune response from the host. This is due to the stimulation of the innate immune cells by glucorhamnan via the toll-like receptor 4. In addition, glucorhamnan could increase the secretion of tumor-necrosis factor alpha, which acts as an inflammatory cytokine [58]. Even though this proinflammatory action has been associated with *Ruminococcus gnavus* [58], it could also be a characteristic of other species, belonging to this genus and could interact with the gut-brain axis.

*Ruminococcus* was reported to be significantly different in a total of ten of the studies. However, the number of times when *Ruminococcus* was more abundant and the number of times, when it was less abundant in ASD children were equal [14,16,17,19,33,34,35,44,51,59]. Regarding its role, there are conflicting findings. *Ruminococcus* was increased in ASD children according to a meta-analysis that analyzed the possible correlation between gut microbiome and ASD [27]. However, two other meta-analyses did not report a statistically significant difference between ASD children and control patients regarding the genus *Ruminococcus* [25,26]. Conversely, there is a hypothesis suggesting that a reduction in *Ruminococcus* abundance could be linked to autism due to its potential impact on arginine metabolism. It has been postulated that decreased levels of *Ruminococcus* may result in higher concentrations of arginine in the bloodstream, leading to increased production of nitric oxide, which could act as a neurotoxin [39]. This further emphasizes the variability in the abundance of different genera across various studies. To gain a clearer understanding of the potential role of these genera in ASD, further research is needed, particularly using animal models under rigorous protocols that eliminate confounding environmental factors. This approach would enable a comprehensive characterization of the microbiome’s role in ASD while controlling for other environmental factors.

### 3.9. Streptococcus

*Streptococcus* is a genus of Gram-positive cocci, and some of its species are beneficial while others are associated with infectious diseases [60,61]. *Streptococcus* is linked to the so-called pediatric autoimmune neuropsychiatric disease associated with *Streptococcus* (PANDAS). This disorder presents with the onset of symptoms typical of autistic behavior, usually after a streptococcal infection [61]. It is thought to be caused by an auto-immune response brought on by cross-reactive antibodies against Streptococcal M protein. These antibodies also act against antigens from the central nervous system, and this is the reason for the observed neuropsychiatric symptoms [61]. Therefore, one could assume that *Streptococcus* could be involved in the etiology of ASD by triggering an abnormal immune response via the gut-brain axis.

This genus was significantly different between ASD and typically developing children in a total of seven of the studies, and its abundance was decreased in the majority of the studies [3,14,33,35,38,39,51]. This finding was in agreement with a published meta-analysis on this subject [26]. The decreased abundance of this genus may be unexpected, based on the assumption that PANDAS is triggered by a *Streptococcus* infection [61]. However, some species belonging to this genus are actually beneficial for their host, for example, *Streptococcus thermophilus*. The last one synthesizes lactic acid and acetate which protect the host by suppressing the growth of harmful bacteria such as *Escherichia coli* [62]. A reduction in the abundance of *Streptococcus* in the gastrointestinal tract (GIT) can compromise its immune-protective functions, allowing other bacterial genera to thrive and potentially influencing the gut-brain axis. Moreover, the presence of *Streptococcus* in the intestines has been found to be negatively associated with fructose intake [60]. Given that children with autism spectrum disorder often exhibit selective eating habits and restricted dietary patterns, their food intake can impact the composition of the gut microbiome [63]. Consequently, in order to validate the potential decrease in *Streptococcus* as an environmental factor in the etiology of ASD, future case-control studies should employ consistent exclusion and inclusion criteria, thus minimizing the influence of dietary factors and ensuring replicable results.

### 3.10. Blautia

The genus *Blautia* is composed of Gram-positive bacteria that are thought to be commensal organisms in the human GIT [64]. *Blautia* has a positive impact on the host’s wellbeing, i.e., it participates in the degradation and metabolism of various substances in the GIT, it synthesizes bacteriocins that have a protective effect, and it is negatively associated with obesity [64]. Therefore, the presence of *Blautia* species in the GIT could be considered to be beneficial to the host.

Our results show that the genus *Blautia* was significantly different in a total of seven of the studies, and its abundance was mostly decreased in ASD children compared to healthy subjects [16,22,23,31,33,48,50]. This was in agreement with a published meta-analysis on the subject of the gut microbiome of autistic children [25]. This finding could be associated with the participation of *Blautia* in the synthesis of serotonin from the gastrointestinal mucosa by stimulating the expression of Tph1 in the gut. Thus, if *Blautia* is decreased, there is also decreased production of serotonin in the GIT [65] and *Blautia* will directly act on the gut-brain axis since serotonin is an important neurotransmitter. Moreover, *Blautia* has a protective function because it can decrease the abundance of pathogenic bacteria in the GIT [64]. As a result of all of the above, a decrease in the genus *Blautia* could be associated with ASD.

## 4. Discussion

Approximately 100 of every 10,000 persons worldwide are affected by autism spectrum disorder, and about one-third of those cases also involve intellectual disability [66]. ASD has been described as a global health issue, whose prevalence could increase even more in the following years [66]. The identification of significantly distinct markers that can be used to screen or treat ASD patients is, therefore, of the utmost importance and the gut microbiome could have a central part in this process.

The results from the present mini review highlight the top ten genera that were identified most times in the screened studies as significantly different between the studied patients and the control subjects. The concept that different compositions of microbiota are related to ASD is not a new one. The hippocampus of germ-free mice has been demonstrated to have abnormal protein and gene expression patterns, as well as alterations in the function of the amygdala [67,68]. After normal microbiota were introduced to the gut, these anomalies were restored [69,70]. Therefore, it is crucial to understand the role of the gut microbiome as a possible factor in the etiology of ASD as it could be easily restored to normal with the help of pre, pro-, and synbiotics, and fecal transplantation therapy [7].

However, the present review has certain limitations. First of all, we only included information about the ten genera that were shown to be most often statistically different. Nevertheless, there are also other genera for which there are data that indicate they are risk factors for ASD, such as Escherichia and Shigella [17,18,44]. However, due to the vast number of potential genera, we only included those for which there were more investigations. In addition, most of the studies involved small cohorts of participants and the reported results had not been replicated in larger trials. Additionally, a number of studies indicated that individuals with ASD may have higher or lower concentrations of the same genus of bacteria. For example, the levels of Bacteroides were increased in some of the studies [14,15,18,19,20,24] and decreased in other studies [3,13,16,17,21,22,23]. These results serve as an illustration of how the presence of the same genus differs among investigations. This could be explained by various external factors that impact the concentration level of the bacteria, but are not always considered when recruiting the participants, i.e., age and comorbidities. There have been reports that the concentration level of Bacteroides decreases with age [17], so the median age of the participants should be close in order to compare the findings of other research groups. Moreover, the concentration levels of different genera could be influenced by diarrhea and constipation and these should be considered when including patients in a study [71,72]. For example, it has been reported that patients with functional constipation had higher levels of Bacteroides [71] and that other comorbidities, such as gall stones, could alter the composition of the microbiome [72]. All of the above-mentioned data show that environmental influences should be taken into consideration because the gut microbiome is particularly sensitive to them. Additionally, eating behaviors may possibly contribute to the changed microbiota of ASD patients [73]. Therefore, a thorough questionnaire examining participants’ diets should also be included in a microbiome investigation.

## 5. Conclusions

The findings of this study demonstrate that the abundance of various bacterial genera within the gut microbiome of children with ASD varies across different studies. Despite significant differences observed, the same genera exhibited decreased levels in some studies while increased levels were reported in other studies. The variations in findings may be attributed to diverse factors such as dietary variations, eating habits, lifestyles, and genetic and environmental influences, all of which can impact the composition of the gut microbiome. To further understand the role of the gut microbiome as a potential etiological component for ASD, interventional and experimental research that involves animal models and eliminates the roles of the environmental factors must be conducted. The results from these studies could serve as a foundation for novel therapeutic strategies such as fecal transplantation therapy. Therefore, it is of utmost importance to conduct research on the intestinal flora under standardized conditions to guarantee reproducibility of results and to effectively control for external factors.

## Figures and Tables

**Figure 1 children-10-01103-f001:**
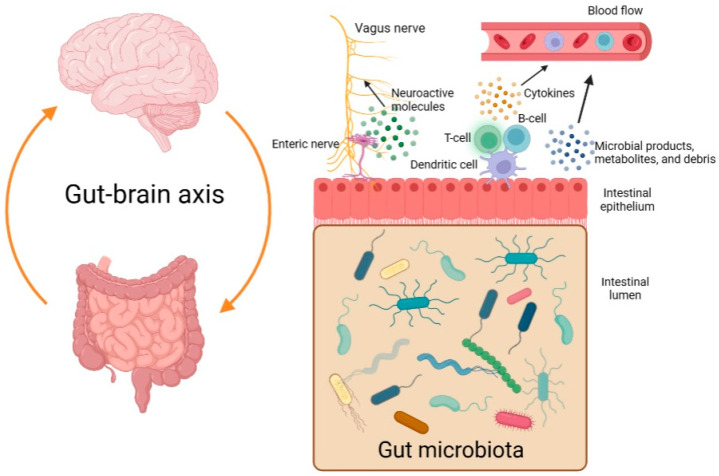
The gut-brain axis and the bidirectional communication between the gastrointestinal tract and the central nervous system (created with BioRender.com, accessed on 6 May 2023).

**Figure 2 children-10-01103-f002:**
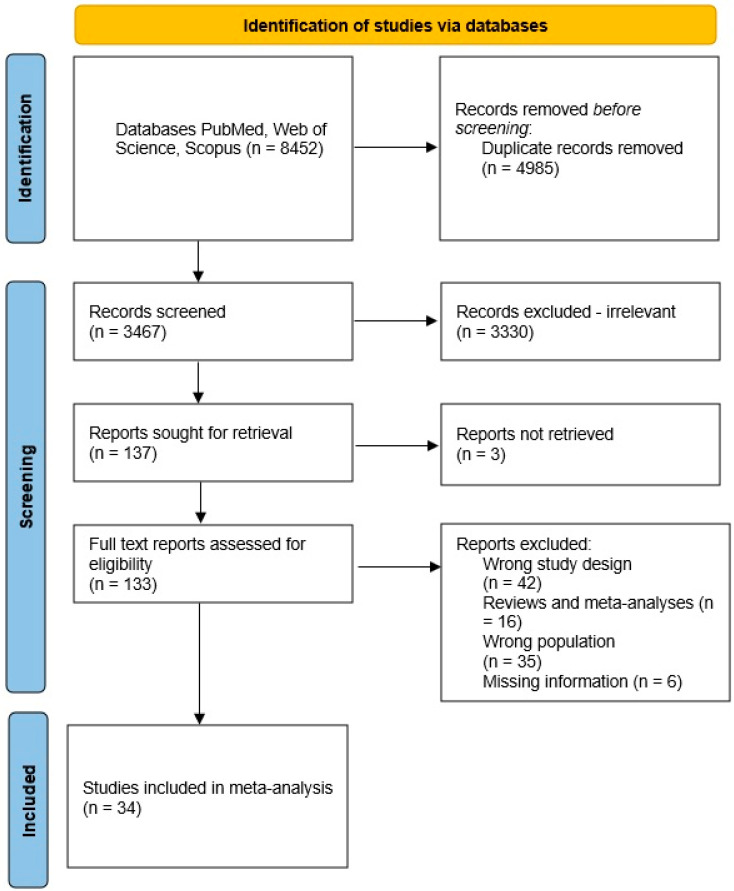
Flowchart of the study selection process.

## Data Availability

No new data were created during this study.

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
