# Peer review of "Genus-Level Analysis of Gut Microbiota in Children with Autism Spectrum Disorder: A Mini Review"

_children, 2023, doi:10.3390/children10071103_

Round 1

Reviewer 1 Report

The ain of this mini-review was to list the bacterial genera for which there is the greatest evidence of a substantial difference between patients with autism specter disorder and typically developing children.
I recommend you organize the article in: introduction, material and methods, results, discussion and conclusions (for example, lines 55-76 withouț lines 67-71 represent material and methods). Limitations of the study should also be discussed at the discussions.
The figures are clear and correct, but I recommend that for figure 1 you specify the source of the image or specify the program in which it was made.
The references are appropriate, the article presents 62 references, being up to date.
Please use the justified alignment.

Author Response

Dear editorial team, dear reviewers, 

Thank you for your letter. We appreciate the interest that the editors and reviewers have taken in our manuscript and the constructive criticism they have given. Please consider our revised manuscript, “Genus-level analysis of gut microbiota in children with autism specter disorder: a mini- review", children-2439447, for publication in Children.  We have addressed the major concerns of the reviewers and we submit a list of changes. All of the changes to the manuscript are indicated with Tracked changes.

Thank you again for consideration of our revised manuscript.

Best regards,

The authors

Reviewer 1

I recommend you organize the article in: introduction, material and methods, results, discussion and conclusions (for example, lines 55-76 without lines 67-71 represent material and methods). Limitations of the study should also be discussed at the discussions. – Thank you for your suggestion. We have reorganized the manuscript and included limitations of the study.

The figures are clear and correct, but I recommend that for figure 1 you specify the source of the image or specify the program in which it was made. – Thank you for your comment. We added the name of the program, used to create the figure.

The references are appropriate, the article presents 62 references, being up to date. – Thank you for your comment.

Please use the justified alignment. – We used justified alignment.

Reviewer 2 Report

This review article is clear and informative. The authors provide a concise overview of Autism Spectrum Disorders (ASD) and the current state of research into their etiology. They highlight the growing interest in the relationship between ASD and the gut microbiota, and the potential significance of this relationship as an etiological component of ASD. Yet, there are some suggestions for improvement, 

1.     The authors could provide more context on the current state of research into the etiology of ASD. While they mention that there has been a lot of research in the last few years, they could provide more information on the key theories and findings in this area.

2.     The authors could provide more detail on the potential significance of the gut microbiota in the etiology of ASD. While they mention that there is growing interest in this relationship, they could provide more information on the mechanisms by which the gut microbiota may influence ASD, and the potential implications for treatment and management of the disorder.

3.     The authors could be more specific in their explanation of the contradictory findings regarding Bacteroides in ASD. While the authors mention that differences among the studied patients could explain the conflicting results, they could provide more detail on the specific patient characteristics that may have influenced the results, such as constipation or diarrhea. Additionally, the authors could provide more detail on the potential implications of these conflicting findings for the role of Bacteroides in ASD, and the need for further research to clarify their role. It also applied to other genera reported in the present study.

4.     The authors could provide more detail on the limitations of the studies included in the review. While they mention that the majority of the research done so far has been observational, they could provide more information on the potential biases and confounding factors that may have influenced the results of these studies.

Minor editing of English language would further enhance the readability of the article. 

Author Response

Dear editorial team, dear reviewers,

Thank you for your letter. We appreciate the interest that the editors and reviewers have taken in our manuscript and the constructive criticism they have given. Please consider our revised manuscript, “Genus-level analysis of gut microbiota in children with autism specter disorder: a mini- review", children-2439447, for publication in Children.  We have addressed the major concerns of the reviewers and we submit a list of changes. All of the changes to the manuscript are indicated with Tracked changes.

Thank you again for consideration of our revised manuscript.

Best regards,

The authors

Reviewer 2

  1. The authors could provide more context on the current state of research into the etiology of ASD. While they mention that there has been a lot of research in the last few years, they could provide more information on the key theories and findings in this area. – Thank you for your suggestion. We added information about the different theories on the etiology of ASD.
  2. The authors could provide more detail on the potential significance of the gut microbiota in the etiology of ASD. While they mention that there is growing interest in this relationship, they could provide more information on the mechanisms by which the gut microbiota may influence ASD, and the potential implications for treatment and management of the disorder. – Thank you for your comment. We have discussed the possible mechanisms by which the gut microbiota might influence the host in the third paragraph of the Introduction section. We have added information about future therapeutic options in the Discussion section.
  3. The authors could be more specific in their explanation of the contradictory findings regarding Bacteroides in ASD. While the authors mention that differences among the studied patients could explain the conflicting results, they could provide more detail on the specific patient characteristics that may have influenced the results, such as constipation or diarrhea. Additionally, the authors could provide more detail on the potential implications of these conflicting findings for the role of Bacteroides in ASD, and the need for further research to clarify their role. It also applied to other genera reported in the present study. – Thank you for suggestions. We included a comment on these conflicting results and the possible reasons in the Discussion section.
  4. The authors could provide more detail on the limitations of the studies included in the review. While they mention that the majority of the research done so far has been observational, they could provide more information on the potential biases and confounding factors that may have influenced the results of these studies. – Thank you for your comment. We have discussed the limitations of our study in the Discussion section.

Minor editing of English language would further enhance the readability of the article. – Thank you for your comment. We have revised the manuscript.

Reviewer 3 Report

I appreciate the opportunity to review this article. I consider that the subject to be dealt with is relevant and deserves study, even more so when dealing with a group of pathologies such as ASD, which are very little studied.

Here are some observations that I think are important to address:

It is necessary to better write the second paragraph in the introduction section. A claim is made that the causes of ASD are unknown, but immediately afterward it is stated that 40% of cases have a specific cause in single gene mutations, and in the next statement it is mentioned that no single gene has been identified as the cause of ASD. Add the quotes in reference to the statements on lines 31 and 32.

References are not placed for various statements in several parts of the text; for example, citations are not placed for the statements made from lines 46 to 54. Please review the entire text in this regard.

After the introduction, I consider it necessary to broaden the importance of the different bacterial genera in the intestinal microbiota, as well as to mention some of those that have been considered relevant in the international literature, to give a preamble to the analysis by genus that is made later.

It is mentioned in the article that a cross-tab analysis is carried out; however, the results are not shown. Please add the results to the text.

Finally, I believe that the conclusion of the review should be broader and include the benefits that knowing this association of the intestinal microbiota with ASD would bring and what types of studies could be considered in the future. Also, consider removing the word "mini" from the article title.

English is good

Author Response

Dear editorial team, dear reviewers,

Thank you for your letter. We appreciate the interest that the editors and reviewers have taken in our manuscript and the constructive criticism they have given. Please consider our revised manuscript, “Genus-level analysis of gut microbiota in children with autism specter disorder: a mini- review", children-2439447, for publication in Children.  We have addressed the major concerns of the reviewers and we submit a list of changes. All of the changes to the manuscript are indicated with Tracked changes.

Thank you again for consideration of our revised manuscript.

Best regards,

The authors

Reviewer 3

It is necessary to better write the second paragraph in the introduction section. A claim is made that the causes of ASD are unknown, but immediately afterward it is stated that 40% of cases have a specific cause in single gene mutations, and in the next statement it is mentioned that no single gene has been identified as the cause of ASD. Add the quotes in reference to the statements on lines 31 and 32. – Thank you for your comment. We rewrote the second paragraph and included quotes for these statements.

References are not placed for various statements in several parts of the text; for example, citations are not placed for the statements made from lines 46 to 54. Please review the entire text in this regard. – Thank you for your comment. We added references to the lines you mentioned and also to other parts of the text.

After the introduction, I consider it necessary to broaden the importance of the different bacterial genera in the intestinal microbiota, as well as to mention some of those that have been considered relevant in the international literature, to give a preamble to the analysis by genus that is made later. – Thank you for your comment. We have added some examples of potentially significant bacteria to the Introduction section.

It is mentioned in the article that a cross-tab analysis is carried out; however, the results are not shown. Please add the results to the text. – Thank you for your suggestion. This type of analysis yields a table of results instead of a chart, which simply lists the frequency with which each species is referenced. Nothing novel would be added to the manuscript as a result of this. Moreover, the same information is available with more details in the supplementary table.

Finally, I believe that the conclusion of the review should be broader and include the benefits that knowing this association of the intestinal microbiota with ASD would bring and what types of studies could be considered in the future. Also, consider removing the word "mini" from the article title. – Thank you for your suggestion. We added new information to the conclusion. As for the word mini, we used ii to indicate that this review is not a systematic one and more information about different genera could be added as well. We chose the top ten genera, for which there were more investigations at the time.

English is good. – Thank you.